# Classification of Drivers’ Mental Workload Levels: Comparison of Machine Learning Methods Based on ECG and Infrared Thermal Signals

**DOI:** 10.3390/s22197300

**Published:** 2022-09-26

**Authors:** Daniela Cardone, David Perpetuini, Chiara Filippini, Lorenza Mancini, Sergio Nocco, Michele Tritto, Sergio Rinella, Alberto Giacobbe, Giorgio Fallica, Fabrizio Ricci, Sabina Gallina, Arcangelo Merla

**Affiliations:** 1Department of Engineering and Geology, University G. d’Annunzio of Chieti-Pescara, 65127 Pescara, Italy; 2Department of Neurosciences, Imaging and Clinical Sciences, University G. d’Annunzio of Chieti-Pescara, 66100 Chieti, Italy; 3Next2U s.r.l., 65127 Pescara, Italy; 4Physiology Section, Department of Biomedical and Biotechnological Sciences, University of Catania, 95123 Catania, Italy; 5National Interuniversity Consortium of Science and Technology of Materials (INSTM), University of Messina, 98122 Messina, Italy

**Keywords:** mental workload, driver monitoring, ADAS, infrared imaging, automotive ergonomics

## Abstract

Mental workload (MW) represents the amount of brain resources required to perform concurrent tasks. The evaluation of MW is of paramount importance for Advanced Driver-Assistance Systems, given its correlation with traffic accidents risk. In the present research, two cognitive tests (Digit Span Test—DST and Ray Auditory Verbal Learning Test—RAVLT) were administered to participants while driving in a simulated environment. The tests were chosen to investigate the drivers’ response to predefined levels of cognitive load to categorize the classes of MW. Infrared (IR) thermal imaging concurrently with heart rate variability (HRV) were used to obtain features related to the psychophysiology of the subjects, in order to feed machine learning (ML) classifiers. Six categories of models have been compared basing on unimodal IR/unimodal HRV/multimodal IR + HRV features. The best classifier performances were reached by the multimodal IR + HRV features-based classifiers (DST: accuracy = 73.1%, sensitivity = 0.71, specificity = 0.69; RAVLT: accuracy = 75.0%, average sensitivity = 0.75, average specificity = 0.87). The unimodal IR features based classifiers revealed high performances as well (DST: accuracy = 73.1%, sensitivity = 0.73, specificity = 0.73; RAVLT: accuracy = 71.1%, average sensitivity = 0.71, average specificity = 0.85). These results demonstrated the possibility to assess drivers’ MW levels with high accuracy, also using a completely non-contact and non-invasive technique alone, representing a key advancement with respect to the state of the art in traffic accident prevention.

## 1. Introduction

Road accidents, indicated as one of the main causes of injury and death, are frequently related to the underestimation of drivers’ mental workload (MW) and fatigue [1]. The world of research is consistent in assuming that crash risks are strongly related to driver mental workload [2,3]. Hence, predicting cognitive states, such as mental overload, could be fundamental to prevent traffic accidents. The quantitative assessment of MW can be performed by means of neuroimaging and neurophysiological techniques and methods [4,5]. Indeed, several studies reported the use of behavioral measurement, such as eye blinking [6,7], and physiological measurement, such as Electrocardiogram (ECG) [8], Electroencephalogram (EEG) [9,10,11], and functional Near Infrared Spectroscopy (fNIRS) [6,12] to estimate MW. In a very recent review on the assessment of MW relying on physiological parameters, Tao et al. stated that cardiovascular, eye movement and EEG measures were the most frequently used across various research fields, reporting 76%, 66%, and 71% of the times a significant association with MW, respectively [13]. Relative to the ECG signal, among all the other physiological parameters, it can be considered one of the most suitable signals in the automotive research domain, since its detection ensures comfort and not excessive invasiveness for the driver if compared, for instance, with EEG measurements. Furthermore, ECG-derived parameters can also be established through cutting-edge technologies, which are now available to a large part of the population (i.e., smart devices) with a good level of reliability [14]. In a recent study, Tjolleng et al., developed a three-level classifier based on artificial neural networks relying on six ECG-derived features extracted in time and frequency domains [15]. In this study, drivers were asked to perform N-back tasks while driving on a static simulator. The developed model reached an accuracy of 82%. However, in general, there is a wide scientific literature about the relationship between physiological parameters (especially HRV) and MW. We recommend a very exhaustive review by Dias et al. [16].

However, the limitations in the assessment of behavioral/physiological parameters in real life driving through the above-mentioned techniques (contact probes, high sensitivity to driver’s motion, specific lighting conditions) prevent their large application in Advanced Driver Assistance Systems (ADAS), in which the use of non-contact sensors would be specifically desirable, and the main aim of the current study is to overtake these limitations due to contact and invasive measurement techniques by the use of a non-invasive and contactless methodology, the thermal infrared (IR) imaging, that has been proposed as a suitable alternative tool to estimate MW, just because of its contactless modality. IR imaging is a non-invasive technology that is able to infer the autonomic modulation of the superficial skin temperature [17]. Importantly, compared with visible cameras used to infer behavioral parameters, IR is not affected by illumination and can work in a completely dark environment. The use of IR imaging permits the estimation of the peripheral autonomic activity relying on the modulation of the skin temperature, which is a known expression of the psycho-physiological state of the subject [18,19,20]. Accordingly, experienced emotions, including stress or fatigue, can produce measurable changes in skin temperature [21,22].

## 2. Related Work

There is great attention in the research field on MW monitoring using thermal IR imaging. Kang et al. assessed affective training times by monitoring the cognitive load relying on facial temperature changes. Significant correlations (i.e., r∈ℝ0.88,0.96) were found between the nose tip temperature and response time, accuracy, and the Modified Cooper Harper Scale ratings [23]. Stemberger et al. proposed a system for the evaluation of MW levels of aviators relying on the assessment of facial skin temperature. The method also relied on head pose estimation, measurement of the temperature variation over different facial regions, and an artificial neural network classifier. The system classified with good accuracy the MW into high, medium, and low levels 81% of the time [24].

Given the advantages of the use of IR imaging in psycho-physiological state monitoring, a relevant number of scientific works on the automotive research field are available. Most of these publications concern driver drowsiness/fatigue monitoring and emotional state detection [25,26,27,28,29,30]. Relative to drivers’ MW monitoring using thermal IR imaging, the literature is instead scarce. Or and Duffy used thermography to assess the relationship between MW and thermal patterns of facial regions of interest (ROIs). They found a significant correlation between the nose skin temperature change and the subjective workload score in both simulated and real-vehicle driving [31]. Pavlidis et al. [32], investigated the effects of cognitive, sensorimotor, emotional, and mixed stressors on driver arousal and performance during a driving simulator experiment. Perinasal perspiration, inferred by IR imaging, together with the measurement of steering angle and the lane departures on the left and right side of the road, revealed a more dangerous driving condition for both sensorimotor and mixed stressors compared to the baseline situation [32]. In a more recent work by Wang et al. [33], the correlation of facial skin temperature and its variation with the EEG-measured MW was examined in three different thermal environments (slightly cool, neutral, and slightly warm). They found that the absolute facial temperature had stronger correlations with MW than facial temperature variation and that the correlations were higher in the neutral thermal environment if compared with the other two thermal conditions [33]. Finally, Perpetuini et al. [12], developed a two-levels Support Vector Machine (SVM) classifier to predict the level of MW from IR imaging features. The Sample Entropy of the fNIRS signal was assumed to indicate MW and was used as output data for the model. The classifier showed a sensitivity of 77% and specificity of 69% [12].

Table 1 summarizes the approaches used in the related work cited above, reporting information for each one about the employed methodology and performance.

In the present work, the driver MW was established by means of IR imaging and supervised machine learning (ML) methods. Supervised ML approaches are part of Artificial Intelligence (AI) algorithms, able to automatically learn functions that map an input to an output based on known input–output pairs (training dataset). The function is inferred from labeled training data and can be used for mapping new datasets (test data), thus permitting to evaluate the accuracy of the learned function and estimate the level of generalization of the applied model [35].

Based on key features of thermal signals extracted from peculiar ROIs indicative of the psycho-physiological state and ECG derived parameters, ML-based classification models of MW were performed with the aim to distinguish among different levels of MW. Two cognitive tests, with their own subcategories, were chosen to investigate the drivers’ response to different and predefined levels of cognitive load in order to categorize the classes of MW. To develop an accurate and automated MW classification system, ML multimodal (based on both IR imaging and ECG derived features) and unimodal (IR imaging or ECG derived features) models were developed and compared. The principal innovation of the present study consists in the capability of distinguishing different MW levels based on the only monitoring of IR signals and/or ECG derived features. Of note, this work describes a novel approach for a contactless methodology dedicated to driver MW classification, constituting a significant improvement to actual ADAS technology and, in general, to road security level. Furthermore, the developed systems can be completely inherited from any other field of application in which it is desirable to accurately define the level of human MW, thus opening new opportunities and perspectives in the domains of ergonomics and human-machine interaction.

## 3. Materials and Methods

### 3.1. Participants

The experimental session involved 26 adults (17 males, age range 18–42, mean 30.89, standard deviation 6.08). Prior to the experimental sessions, the participants were adequately informed about the purpose and protocol of the study, and they signed an informed consent form resuming the methods and the purposes of the experimentation in accordance with the Declaration of Helsinki [36].

Participants were selected according to the following inclusion criteria:-possession of a driver’s license;-aged 18 years old or over;

Participants were excluded if they did not fall under the inclusion conditions and if they were diagnosed with mental/cognitive impairment.

A survey conducted through the administration of questionnaires revealed that, on average, participants had an experience of driving for (16.54 ± 5.84) years, they were used to driving (53.64 ± 18.45) hours per day and (6.29 ± 1.13) days per week. Furthermore, 72.73% declared that they drive only in an urban context, and 9.09% mainly on highways, whereas the 18.18% declared mixed context driving.

### 3.2. Experimental Protocol

Prior to testing, each subject was left in the experimental room for fifteen minutes to allow their baseline skin temperature to stabilize. The environmental conditions of the experimental room were set at a standardized temperature (23 °C) and humidity (50–60%) by the use of a thermostat.

The experimental sessions were performed using a static driver simulator (Figure 1a). Three 27 inch monitors were used to display the scenario, with a total video resolution of 5760 × 1080 pixels. The distance between the driver and monitors was 1.5 m. Drivers’ horizontal view angle was 150 degrees. Participants sat comfortably on the driver’ seat during both acclimatization and experimental periods.

The software used for the driving simulation was City Car Driving, Home Edition software-version 1.5 [37] (Figure 1b). The experimental protocol consisted in performing a 45 min driving simulation in an urban context. The experimental conditions were set a priori to ensure adverse driving conditions and deliver a reproducible experimental protocol to all study participants (i.e., Traffic density: 60%; Traffic behavior: Intense traffic; Pedestrian crossing the road in a wrong place: Often; Dangerous change of traffic: Often; Emergency braking of the car ahead: Often).

These conditions represented the baseline (BL) situation for the drivers and were selected to guarantee a non-monotonous environment. In particular, the settings associated to emergency situations and traffic guaranteed uncomfortable driving, since the participants were often driving in non-monotonous situations. After a BL period of fifteen minutes, two cognitive tests (i.e., Digit Span and Rey Auditory Verbal Learning tests) were administered to drivers. The administration of cognitive tasks allowed to manipulate the MW with respect to the baseline driving.

In detail, the Digit Span test (DST) is a cognitive test composed of two different tasks able to assess the abilities of short-term memory and working memory, the latter referring to the skill to retain information for a short time to manipulate them mentally [38]. For this test, the participant was asked to repeat sequences of digits verbally presented with a pace of one digit per second. The test started with a two-digit series and each time the sequence was repeated correctly a new set was presented with one more digit. If the participant could not remember a series, another one of the same length was proposed. If the participant was not able to repeat two sequences of the same length, the test ended. The digit span score consisted in the length of the longest correctly recalled sequence. For the purposes of the present study, the DST was composed of two parts:Forward DST (repetition of digits in the same order to their presentation)Backward DST (repetition of digits in the reverse order to their presentation)

The Rey Auditory Verbal Learning test (RAVLT) is a cognitive task able to evaluate verbal learning and long-term memory [39,40]. It was administered by reading the participant a list of 15 words at the pace of one word per second. At the end of the reading, the subject was asked to immediately repeat as many words as possible, in any order. This procedure was repeated with the same word list five consecutive times, recording different elements each time. This was the first part of the test and consisted in the Immediate Recall (ImmR).

After a 15-min time interval, during which the subject continued to drive, he/she was asked to remember (without the list being re-proposed by the examiner) as many words as possible from the list. This was the second part of the RAVLT and consisted in the Delayed Recall (DelR).

Subsequently, the last part of the test consisted in the Recognition (Rec) of the 15 words among the other words not present in the original list. The total amount of items was 46. This test allows for a qualitative evaluation of the memory performance in terms of facilitated recovery.

At the end of the test, the subject drove for 10 min without further test administration. The whole pipeline of the experimental procedure is summarized in Figure 2:

### 3.3. Data Acquisition and Analysis

During the execution of the experimental protocol, ECG signals and visible and thermal IR videos were concurrently acquired.

The ECG signals were recorded by means of Encephalan Mini (Medicom MTD system, Taganrog, Russia) using the lead configuration determined by the Standard Limb Leads (i.e., electrodes positioned at the right arm (RA), left arm (LA), and left leg (LL)) [41]. The ECG signals were acquired at a frequency rate of 256 Hz and band-pass filtered in the frequency band of [0.05–150] Hz. Furthermore a notch filter was used to eliminate the artifact due to the mains power supply (f_notch_ = 50 Hz).

An Intel RealSense D415 depth camera (Intel Corporation©, Santa Clara, California, USA) and a FLIR Boson 320LW IR thermal camera (FLIR corporation©, Wilsonville, Oregon, USA) were used to acquire visible and thermal IR videos, respectively. In detail, the visible camera is Full HD 1080p (1920 × 1080 pixel), whereas the spatial resolution of the thermal camera is 320 × 256 pixels. Relative to the thermal camera, FLIR Boson 320 relies on uncooled VOx microbolometer technology and it is featured with a thermal sensitivity of 50 mK. Of note, the output of FLIR Boson 320 is a 16 bit signal, linear with input flux (i.e., target irradiance) and independent from the camera’s temperature. This means that the output is not translated to absolute temperature (i.e., K/°C), and it ranges from 0 to 216.

For the purposes of this study, the two imaging devices were held together and aligned horizontally by means of a specifically designed frame made by Next2U^®^ (Figure 3a,b).

Both visible and IR videos were recorded at a frequency rate of 10 Hz. The distance between the participant and the imaging system was 0.6 m (Figure 1a).

Concerning IR imaging data analysis, visible imagery were used to track facial landmarks (i.e., 68 points) through the software OpenFace [42], and, successively, they were co-registered to the thermal imagery by the estimation of the geometrical transformation between the visible and the IR optics, following the same procedure described in [43] (Figure 4a). Two ROIs were automatically determined on facial areas of physiological importance (i.e., nose tip and glabella) (Figure 4b). For each ROI, the average value of the pixels was extracted over time and representative features were computed over each experimental phase. To remove possible artifacts from thermal signals, the Hampel function (MATLAB 2021b©, Mathworks Natick, MA, USA) was employed [44]. The Hampel filter is a robust outlier detector relying on Median Absolute Deviation. For each sample of the signal, median and standard deviation are calculated using all neighboring values within a window of size SampWin. If the point of interest lies n_SD_ standard deviations from the median it is identified as an outlier and is replaced by the median value. In this work, we chose SampWin = 15 s and n_SD_ = 2. To take into account the initial values of the thermal signals, the average values of baseline (evaluated across a period of one minute before the DST phase) were subtracted from the raw thermal signals for each experimental phase (Figure 4c).

Subsequently, the following features were extracted from the thermal signals:Mean value (MeanTemp);Standard deviation (STD);Kurtosis (K);Skewness (S);90th percentile (90th P);Sample Entropy (SampEn);Ratio of the power spectral density evaluated in the low-frequency band (LF = [0.04–0.15] Hz) and in the high-frequency band (HF = [0.15–0.4] Hz) (LF/HF);Mean value of the power spectral density evaluated in the low-frequency band (LF = [0.04–0.15] Hz);Mean value of the power spectral density evaluated in the high-frequency band (HF = [0.15–0.4] Hz).

For the ECG data analysis, the elapsed time periods between the two successive R-peaks of the ECGs (RR signals) were extracted by means of a home-made MATLAB 2016b© script. The script was based on a peak detection procedure, in which parts of the signal exceeding two standard deviations were considered as R peaks. On the obtained RR signals (i.e., Heart Rate Variability (HRV) signal), six features were computed over the experimental phases:Mean value (RRmean);Standard deviation (SDNN);Root mean square of successive differences (RMSSD);Ratio of the power spectral density evaluated in the low-frequency band (LF = [0.04–0.15] Hz) and in the high-frequency band (HF = [0.15–0.4] Hz) (LF/HF);Mean value of the power spectral density evaluated in the low-frequency band (LF = [0.04–0.15] Hz);Mean value of the power spectral density evaluated in the high-frequency band (HF = [0.15–0.4] Hz).

To take into account the initial values of the HRV signals, each one of the features was normalized with respect to their baseline value. In particular, the ratio between each HRV feature during the experimental phases and the same feature evaluated during the baseline was computed and considered as input for the models.

Each one of the IR and ECG features were extracted relatively to the specific subcategory of test, which were used to define the label classes for the ML based models. For each one of these classes, features have been computed working on the IR or ECG signal acquired during the specific subtest. This aspect guarantees a balanced class numerosity because all the features for all the subjects were considered relative to each of the classes. A detailed description of the features computation is reported in the Appendix A Section.

### 3.4. Application of Supervised Machine Learning for Classification

Supervised ML is the process of learning a set of rules from instances with the aim of automatically find functions that map an input to an output. The function is inferred from labeled training data and can be used for mapping new dataset (test data) thus allowing to evaluate the accuracy of the learned function and estimate the level of generalization of the applied model [45]. In the present study the performances of six categories of classifiers were compared: Decision Trees (DT) [46], Discriminant Analysis (DA) [47], Logistic Regression (LR) [48], Support Vector Machines (SVM) [49], Nearest Neighbor (kNN) [50], and Ensemble Classifiers [51].

Linear, quadratic and cubic SVM classification models were considered in the present work.

Coarse, medium and fine kNN classification models were considered in the present work.

Relatively to the Ensemble classifiers, bagged trees, subspace discriminant, subspace kNN, and Random Under-Sampling (RUS) boosted trees were considered in the present work [52].

A k-fold cross validation (with k = 5) was used to protect against overfitting [53]. The procedure relies on partitioning the dataset into folds, each one with a training and validation dataset, and estimating the accuracy on each fold, guaranteeing the generalization of the model. To ensure that the samples of the same subject would have not been considered in both training and validation procedures, the folds were created so as to ensure that each subject was seen by the model only in the training or in the validation phases, and not in both of them. For the sake of clarity, for each set of features and each classifier model, since the subjects were 26, a set of 21 subjects were employed for training and a set of five drivers were used for testing. The procedure was iteratively repeated, randomizing the subjects involved as training and validation sets.

The machine learning-based analysis of data were performed by means of the Classification Learner App, MATLAB 2021b© [54]. For the purpose of this work, all the classification models available were considered.

## 4. Results

### 4.1. Drivers’ Performances on Cognitive Tasks

The DST score was calculated as the length of the longest correctly recalled sequence in both Forward and Backward phases. The RAVLT scores were calculated counting the total number of words repeated in the five repetitions over the ImmR phase, counting the total number of words recalled by the participants during the DelR phase and the total amount of correctly recognized world during the Rec phase.

Since the label classes of the developed models (Figure 12) were based on the specific subcategories of the DST and RAVLT, it was necessary to objectively assess if the effect on the performances of the test on the cohort of subjects was adherent with the one reported in the literature. Indeed, paired *t*-test analyses were performed on the scores obtained by the participants during the execution of the two cognitive tests. Significant differences were observed between Forward and Backward DST (t = 2.69, *p* < 0.01, degrees of freedom (dof) = 25), between ImmR and DelR (t = 21.90, *p* << 0.01, dof = 25), and between DelR and Rec scores (t = −44.82, *p* << 0.01, dof = 25). No significance was found in the comparison between ImmR and Rec. Participants’ scores are reported in Figure 5 (i.e., whiskers plot).

The mean values and standard deviations of the participants’ scores are reported in Table 2.

### 4.2. IR-Visible Video Processing

The method for combined visible and IR video processing has been validated in [43]. For the present study, on average 95.20% of the video frames were correctly processed. This percentage value referred to the number of frames with correctly identified facial landmarks with reference to the total number of frames.

Regarding the computational load, the average execution time of the developed algorithm was 0.09 s/frames with MATLAB 2016b© (64-bitWindows 7 Pro, Service Pack 1; Intel (R) Core (TM) i5 CPU; 8.00 GB RAM).

### 4.3. Performances of Supervised Machine Learning Approaches

Thermal and HRV features were first investigated and statistical analysis were performed to assess the most informative features among them. Student’s t-tests were performed among all the features over the experimental phases. The results are reported in Figure 6, Figure 7 and Figure 8 for DST and Figure 9, Figure 10 and Figure 11 for RAVLT.

Relative to DST, the skewness and the 90th percentile were the most informative IR features for the nosetip, whereas the LF/HF feature gave an important contribution for both nosetip and glabella (Figure 6 and Figure 7). No significant difference was revealed by HRV derived features between the two experimental phases (Figure 8).

Referring to RAVLT, the skewness, the 90th percentile and LF/HF were the most informative IR features relative to nosetip, showing significant differences in the comparison of ImmR—DelR (Figure 9). Instead, standard deviation and LF/HF were the most informative features relative to glabella for every comparison among the experimental phases, sampEn for ImmR vs. DelR comparison and LF and HF features for the comparison ImmR vs. Rec (Figure 10). For HRV derived features, HF and LF features were the most informative, both showing significant differences in the comparison DelR vs. Rec (Figure 11). HF features also showed a significant difference in the comparison ImmR vs. Rec (Figure 11).

In the present work, unimodal and multimodal ML-based approaches were developed, each one of them relying on six categories of classifiers: Decision Trees (DT), Discriminant Analysis (DA), Logistic Regression (LR), Support Vector Machines (SVM), Nearest Neighbor (kNN), and Ensemble Classifiers. In the unimodal approach, features extracted from IR signals or HRV signals were separately used as input signals. In the multimodal approach, instead, features extracted from both IR and HRV signals were used together as input data for the classification models. Two-level and three-level classification models were adopted for DST and RAVLT data, respectively. The scheme of the classification model is reported in Figure 12.

Notably, not all the features were used as input to the classifier models, but they were selected through a wrapper method [55,56]. This feature selection approach allows to consider only the minimal set of features that are relevant for the classification purpose. In particular, the random subset of features are evaluated as input features of the specific model and the subset of features that reach the best performance are chosen as input data. In this study, a number of 50 random combinations of features was chosen. After this procedure, for DST and RAVLT, the best feature sets were available and constituted the effective input data for the classifiers. The set of features after the wrapper procedure are summarized in Table 3.

The results of the classifications, in terms of accuracy, are reported in Table 4.

Relative to the classifiers with the best performances (highlighted in bold in Table 4), the Receiver Operating Characteristics (ROC) curves and confusion matrices are reported in Figure 13 for DST and Figure 14 and Figure 15 for RAVLT. ROC curves represent sensitivity (i.e., true positive rate) versus specificity (i.e., 1-false positive rate) across a range of values to evaluate the ability of the classifier to predict an outcome. An important parameter is the Area Under Curve (AUC), which summarizes the classifier performances. A model whose predictions are 100% wrong has an AUC of 0, whereas a model whose predictions are 100% correct has an AUC of 1.

In Figure 13, the performances of the linear SVM classifiers for both unimodal IR features-based models and the multimodal IR + HRV features-based classifier are reported. The unimodal IR features based model showed good performance (accuracy = 73.1%; AUC = 0.76; sensitivity = 0.73; specificity = 0.73; precision = 0.73; F1-score = 0.73) as well as the multimodal IR + HRV features based classifier (accuracy = 73.1%; AUC = 0.80; sensitivity = 0.69; specificity = 0.71; precision = 0.69; F1-score = 0.72).

Figure 14 shows the performances of the three-level unimodal IR features based classifier relying on the ensemble bagged tree model. Dealing with a three-level classifier, three ROC curve are presented, each one of them representing the comparison of one class with the cumulative class of the other two. The average performances are (AUC = 0.85; sensitivity = 0.71; specificity = 0.85; precision = 0.71; F1-score = 0.70).

Figure 15 shows the performances of the three-level multimodal IR + HRV features based classifier relying on the ensemble bagged tree model. The average performances are (AUC = 0.85; sensitivity = 0.75; specificity = 0.87; precision = 0.75; F1-score = 0.74).

## 5. Discussion

Monitoring the MW during driving situations is of paramount importance given its close relationship with the risk of road accidents. The main aim of the present study was to develop models, based on drivers’ psychophysiological features, that are able to discriminate the level of drivers’ MW. To this specific aim, two different cognitive tests (DST and RAVLT) were administered to twenty-six participants while driving in a simulated environment under non-monotonous situations. The statistical analyses on cognitive tests scores reveled significant differences between Forward and Backward DST and between ImmR and DelR and between ImmR and Rec in RAVLT, thus revealing different performances of subjects over the experimental phases.

DST and RAVLT were specifically chosen to study two different types of cognitive load, the former related to short-term memory and the latter related to long-term memory and verbal learning. DST and RAVLT are indicative also of the working memory capacity (WMC) of the subjects [57]. In particular, the Backward DST and the DelR-RAVLT have been reported as the most demanding in terms of WMC [57]. Estimating the WMC is of crucial importance in the research domain of the automotive sector since it has been demonstrated as a predictor of distracted driving [58]. In this study, the authors demonstrated that the levels of WMC affect the driving performances of individuals while engaged in cognitive distraction. Furthermore, they reported a mediation of WMC on the effect of distraction on braking response time.

In the current study, the possibility of recognizing different levels of MW through non-invasive techniques was investigated, and ML-based models relying on drivers’ psychophysiological features were developed and compared. HRV and IR thermal features were extracted over the experimental phases. In this context, it has to be underlined that the novelty of the present study consists in validating models able to classify different kinds of MW relying on the IR imaging technique, which is a completely non-invasive and contactless methodology. Relative to DST, the most informative IR features were the skewness and the 90th percentile for the nosetip and LF/HF for both nosetip and glabella (Figure 6 and Figure 7). HRV derived features revealed no significant difference (Figure 8). Regarding RAVLT, the most informative IR features relative to nosetip were the skewness, the 90th percentile, and LF/HF, which globally showed significant differences in the comparison of ImmR vs DelR (Figure 9). Instead, the most informative features relative to the glabella were the standard deviation and LF/HF for every comparison among the experimental phases, sampEn for ImmR vs. DelR comparison and LF and HF features for the comparison of ImmR vs. Rec (Figure 10). For HRV derived features, HF and LF features were the most informative, both showing significant differences in the comparison of DelR vs. Rec (Figure 11). HF features also showed a significant difference in the comparison of ImmR vs. Rec (Figure 11).

The relevance of IR features in discriminating different levels of MW is appreciable from the results mentioned above. Specifically, significant features relative to nosetip are commonly involved in both DST and RAVLT, whereas the features related to the glabella region are mostly involved during RAVLT. This result can be due to the major cognitive involvement in RAVLT since it has been demonstrated that thermal signals from the glabella/forehead are directly linked with cognitive load [59,60,61]. An important role is also played by the nosetip thermal features, and this result is in accordance with the literature [1,59,61]. Generally, the nosetip region has been reported as the most responsive during cognitive tasks, reflected by a drop in the nosetip temperature during cognitive task executions with respect to baseline conditions [62]. Of note, LF/HF for both the nosetip and glabella regions was demonstrated to be relevant for assessing the MW level. In fact, this feature accounts for the balance between the sympathetic nervous system (SNS) and the parasympathetic nervous system (PNS) activity. It has been inherited by HRV metrics and it is based on the assumption that LF power is generated by the SNS, while HF power is produced by the PNS [63]. The LF/HF feature has already been used for thermal imaging data analysis, showing good contribution in the psychophysiological state assessment of individuals [25,61].

Regarding the HRV derived features, it has been observed that they revealed statistical significance only in RAVLT, and, in particular, LF and HF features were the most informative. As mentioned above, they are relevant for the activation of SNS and PNS, respectively. Between them, the most informative was the LF feature, especially in the comparison between DelR and Rec. A recent review from Forte et al. reported HF and LF features from HRV as significant features indicative of cognitive performances [64]. However, the scientific community is not commonly in accordance on the fact that HRV can be a reliable indicator of MW, especially in the field of automotive. In fact, Paxion et al., in a review on mental workload and driving, highlighted some limits of HRV indicators [4]. Specifically, they argued that HRV is not exclusively sensitive to changes in MW, but is also related to energetic, thermoregulatory, respiratory, and emotional processes and physical activity. Furthermore, they reported that HRV is not always able to discriminate the level of difficulty, thus being an insufficient indicator to assess the MW.

This kind of finding is indeed reflected from the results of the present study, with reference not only to the most informative features but also in regard to the developed ML models. In this study, several models based on ML approaches have been compared based on unimodal IR/unimodal HRV/multimodal IR + HRV features. As shown in Table 4, the best classifier performances were reached by the unimodal IR features-based classifier and also from multimodal IR + HRV features-based models. Unimodal HRV- based models showed the worst performances with respect to the other approaches (i.e., the overall best accuracies reached were 59.6% and 47.4% for DST and RAVLT, respectively). Of note and with reference to the state of the art in ML and deep learning, in this work a specific typology of multimodal approach has been used. Indeed, referring to the recent work of Guarino et al., together with the unimodal approach defined by the authors also single-view learning, a single typology of multimodal procedure was adopted in the present study, referred to by the authors as the intermediate integration multi-view approach [65]. This particular multi-view approach has been chosen since there was the necessity of having a feature selection step (i.e., wrapper method) in the analysis pipeline, prior to the concatenation of IR and HRV features. Hence, the early integration multi-view approach was not considered, since the number of input features was similar to the number of participants. Further studies, instead, could be done to implement the late integration multi-view approach. In this regard, it is necessary to enlarge the sample size to benefit from more reliable single classifiers. For DST, the best performing models were based on two-classes of SVM classifier with linear kernel with unimodal IR features (accuracy = 73.1%; AUC = 0.76; sensitivity = 0.73; specificity = 0.73) and multimodal IR + HRV features (accuracy = 73.1%; AUC = 0.80; sensitivity = 0.69; specificity = 0.71) (Figure 13). For RAVLT, the best performing models were based on three-classes: ensemble bagged trees classifier with unimodal IR features (accuracy = 71.1%; average AUC = 0.85; average sensitivity = 0.71; average specificity = 0.85) and multimodal IR + HRV features (accuracy = 75.0%; average AUC = 0.85; average sensitivity = 0.75; average specificity = 0.87) (Figure 14). Of note, for RAVLT, three-class SVM classifiers with linear kernels also reported good performances with accuracies of 65.8% and 63.2% for unimodal IR and multimodal IR + HRV features based models (Table 4).

The high performances obtained from unimodal IR features-based classifiers are of paramount importance given the possibility of determining the level of drivers’ MW based on features collected by a non-contact device, i.e., the thermal camera. Thermal IR imaging outperformed the HRV measurements as well, constituting a reliable mean for assessing the level of MW in a ubiquitous and non-contact manner. This is an important result, given that in the automotive domain, especially in ADAS, one of the most important aims is to determine the psychophysiological state of the driver without interfering with him/her to avoid/prevent traffic accidents. The impact of such a result is interesting also in terms on ergonomics applied in the automotive field. In fact, the developed ML model could communicate the cognitive state of the driver and alert him/her in case of moderate/high MW. Furthermore, the results are obtained relying on a small-sized thermal camera (i.e., FLIR Boson 320), highly suitable for applications in a restricted environment, such as the cockpit of a vehicle.

However, some limitations have to be mentioned. First, further studies should be performed to increase the sample numerosity. The ML approaches used in this study relied on supervised learning, which is inherently a data-driven analysis; data-driven analyses are highly affected by the sample numerosity, and the performance of the model could indeed improve, reducing a possible overfitting effect driven by the limited sample size. Moreover, increasing the sample size could open the way to more sophisticated and powerful approaches based on deep learning modeling, which is the state of the art in data analysis in several areas of research.

Second, the current study focused on drivers with a limited age range (i.e., 18–42 years old), involving only young and middle-aged adults. The most important improvement of the method could be obtained, including in the study sample individuals with a wider age range. Furthermore, beyond increasing the sample size and age range, other factors, such as thermal comfort, gender and weather conditions during simulated driving sessions will be considered [66,67,68,69]. In fact, accounting for these factors could be of primary valence in automotive research, leading to a broad overview of all aspects concerning the object of the study.

Moreover, the present results refer to simulated driving conditions in which determinant variables for IR measurements, such as sunlight or forced ventilation, were not considered. Therefore, it would be desirable to also apply the developed methodology on real-driving situations in order to generalize the applicability of the technique.

As for being state-of-the-art, this is an original and novel study concerning drivers’ MW evaluation by means of thermal imaging, employing supervised ML algorithms. The present study, although addressed to limited and specific experimental conditions, underlines the feasibility of the method to be verified under wider operating situations. The present work represents a step forward in the perspective of the prevention of road accidents and, above all, it can constitute a turning point in the identification of various levels of mental workload, with benefits in several research domains, from ergonomics to human machine interaction.

## 6. Conclusions

In the present work, a novel method for drivers’ MW evaluation is presented. In particular, MW levels of the subjects while driving in a simulated environment were estimated with a high level of accuracy through ML algorithms applied to IR and HRV data. The presented work constitutes a step towards the establishment of a reliable detection of the MW levels in a non-invasive and contactless manner, ensuring the maintenance of an ecologic condition of driving, possibly contributing to the prevention of traffic accidents. Further directions for development will include the validation of the developed method directly on-board, with live evaluation of the cognitive workload level of the drivers. This will be particularly useful for all of the categories of long-time drivers, such as truck drivers and bus drivers, in order to prevent traffic accidents due to an excessive cognitive workload during driving activity.

## Figures and Tables

**Figure 1 sensors-22-07300-f001:**
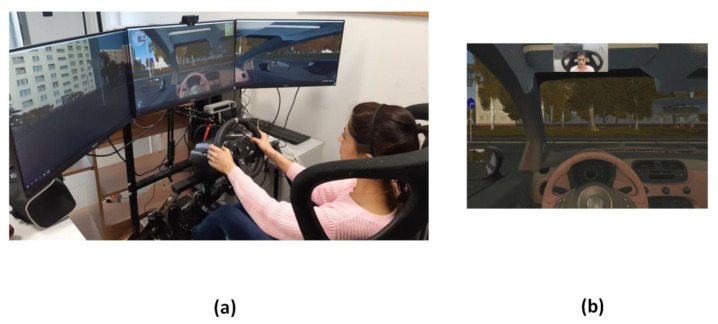
Experimental setting: (**a**) static driver simulator; (**b**) screenshot of the driving simulation software (i.e., City Car Driving, Home Edition software-version 1.5, Forward Development, Ltd., Verona (WI), USA [37]).

**Figure 2 sensors-22-07300-f002:**
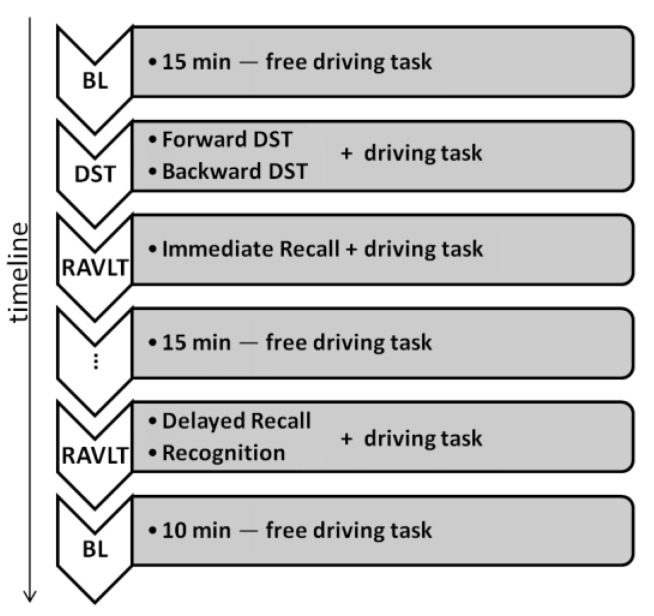
Pipeline of the experimental protocol.

**Figure 3 sensors-22-07300-f003:**
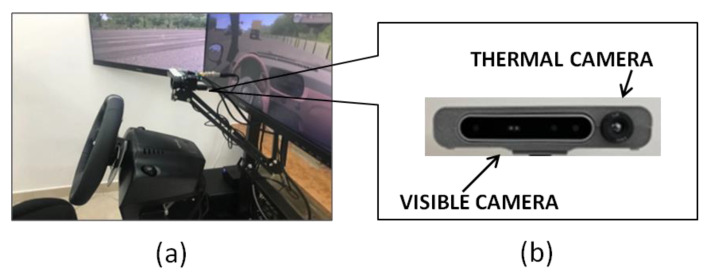
Imaging system device settings: (**a**) position of the imaging system in the driving simulator; (**b**) detail of the imaging system device (visible and thermal camera horizontally aligned and held together by means of a 3d-printed support).

**Figure 4 sensors-22-07300-f004:**
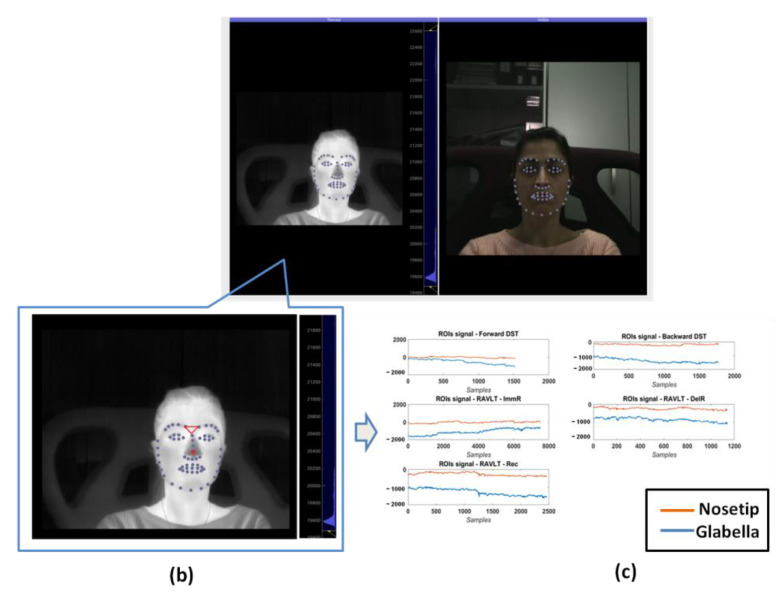
Processing of thermal and visible videos. (**a**) Software interface for the acquisition and processing of visible and thermal IR videos. (**b**) Thermal image with ROI drawn in red colors (Nosetip and Glabella); (**c**) thermal signal extracted from the two ROIs during the experimental phases. The values are obtained subtracting the mean value of the signals during the baseline phase.

**Figure 5 sensors-22-07300-f005:**
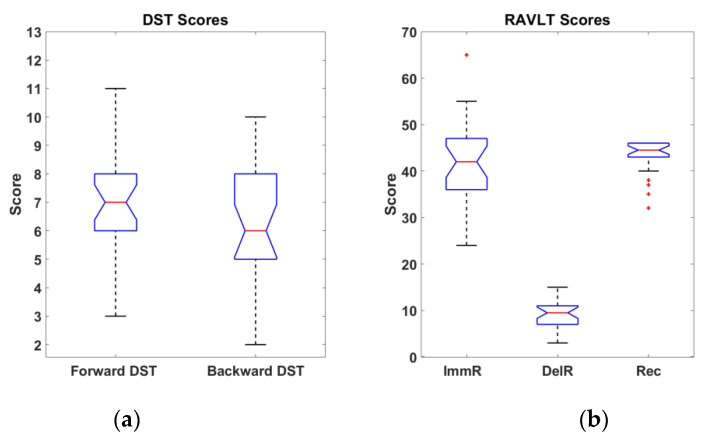
Whiskers plot of the participants’ scores in DST (**a**) and RAVLT (**b**). Outliers are represented with red crosses.

**Figure 6 sensors-22-07300-f006:**
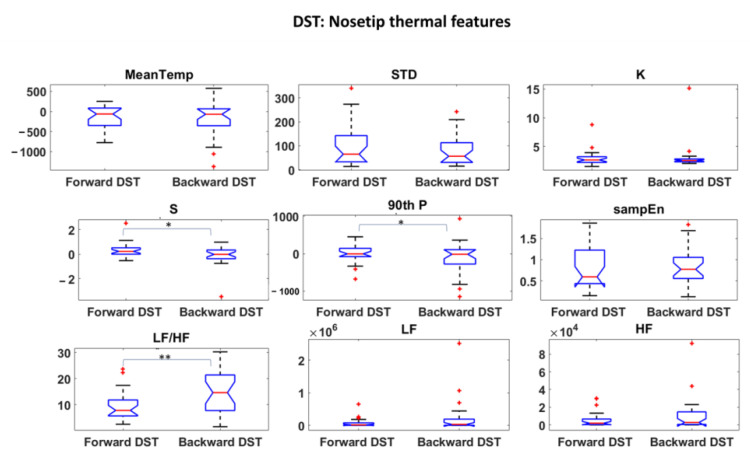
Thermal features relative to Nosetip ROI extracted during DST (* *p* < 0.05; ** *p* < 0.01). Outliers are represented with red crosses. The titles of the single plots refer to abbreviations of features described in Section 3.3.

**Figure 7 sensors-22-07300-f007:**
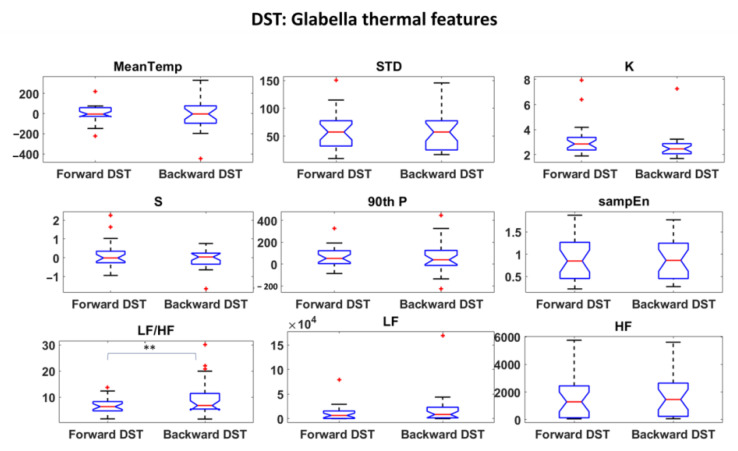
Thermal features relative to Glabella ROI extracted during DST (** *p* < 0.01). Outliers are represented with red crosses. The titles of the single plots refer to abbreviations of features described in Section 3.3.

**Figure 8 sensors-22-07300-f008:**
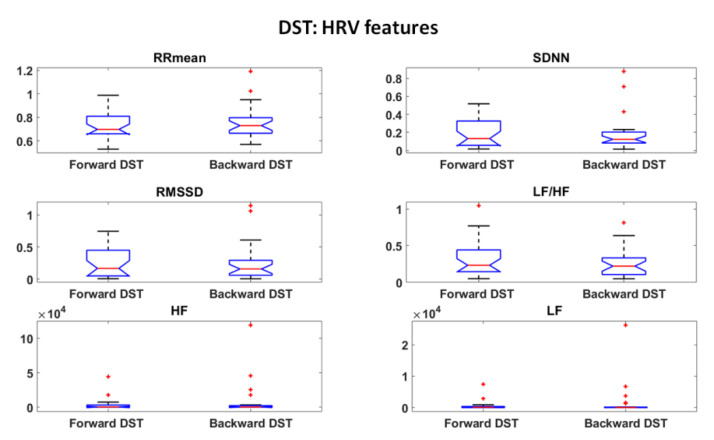
HRV features extracted during DST. Outliers are represented with red crosses. The titles of the single plots refer to abbreviations of features described in Section 3.3.

**Figure 9 sensors-22-07300-f009:**
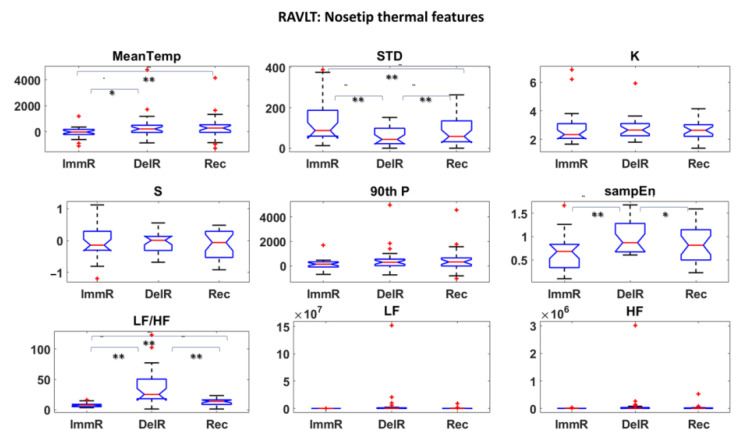
Thermal features relative to Nosetip ROI extracted during RAVLT (* *p* < 0.05; ** *p* < 0.01). Outliers are represented with red crosses. The titles of the single plots refer to abbreviations of features described in Section 3.3.

**Figure 10 sensors-22-07300-f010:**
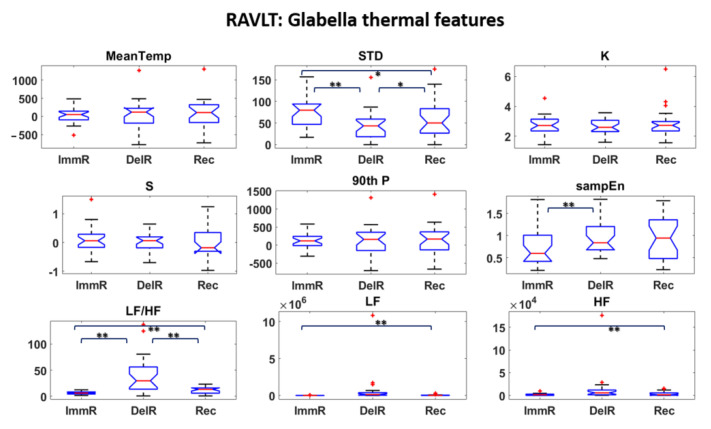
Thermal features relative to Glabella ROI extracted during RAVLT (* *p* < 0.05; ** *p* < 0.01). Outliers are represented with red crosses. The titles of the single plots refer to abbreviations of features described in Section 3.3.

**Figure 11 sensors-22-07300-f011:**
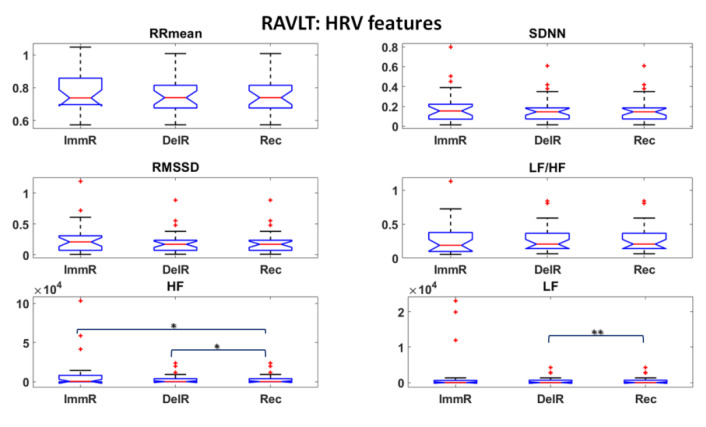
HRV features extracted during RAVLT (* *p* < 0.05; ** *p* < 0.01). Outliers are represented with red crosses. The titles of the single plots refer to abbreviations of features described in Section 3.3.

**Figure 12 sensors-22-07300-f012:**
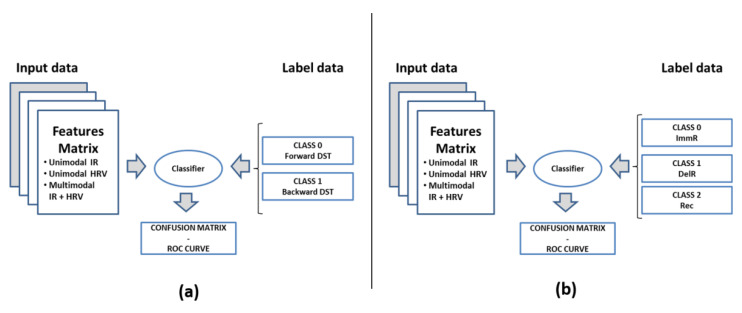
Scheme of classification adopted in the present work: (**a**) scheme of the two-level classification model for DST; (**b**) scheme of the three-level classification model for RAVLT.

**Figure 13 sensors-22-07300-f013:**
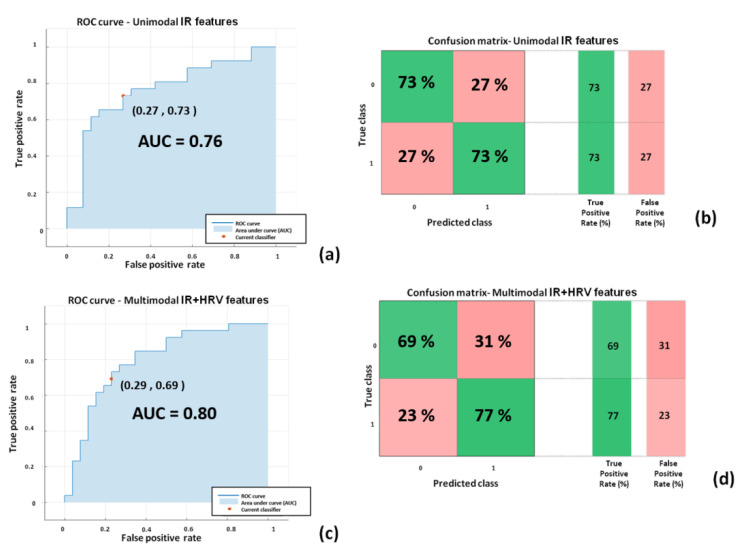
Performances of the SVM classifiers for DST: (**a**) ROC curve for unimodal IR features-based classifier; (**b**) confusion matrix unimodal IR features-based classifier; (**c**) ROC curve for multimodal IR + HRV features-based classifier; (**d**) confusion matrix multimodal IR + HRV features-based classifier.

**Figure 14 sensors-22-07300-f014:**
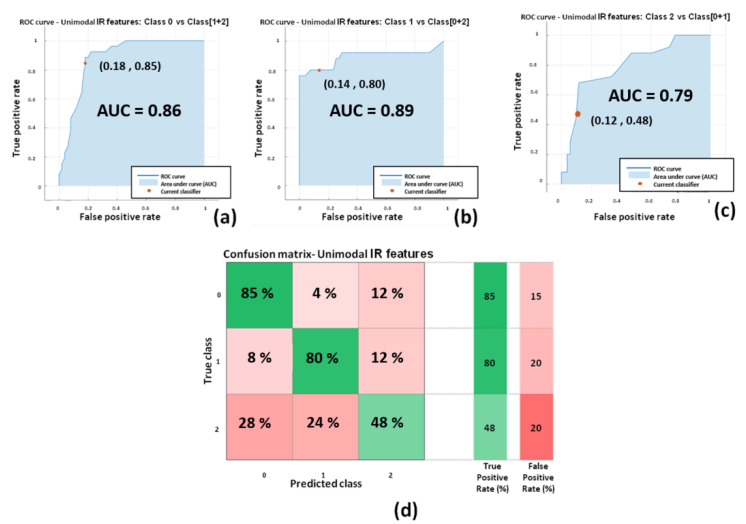
Performances of the Ensemble bagged trees for unimodal IR features-based classifiers for RAVLT: (**a**) ROC curve for the classifier of class ImmR (i.e., class 0) vs. cumulative class (DelR + Rec) (i.e., class 1 + 2); (**b**) ROC curve for the classifier of class DelR (i.e., class 1) vs. cumulative class (ImmR + Rec) (i.e., class 0 + 2); (**c**) ROC curve for the classifier of class Rec (i.e., class 2) vs. cumulative class (ImmR + DelR) (i.e., class 0 + 1); (**d**) confusion matrix for the unimodal IR features-based classifier.

**Figure 15 sensors-22-07300-f015:**
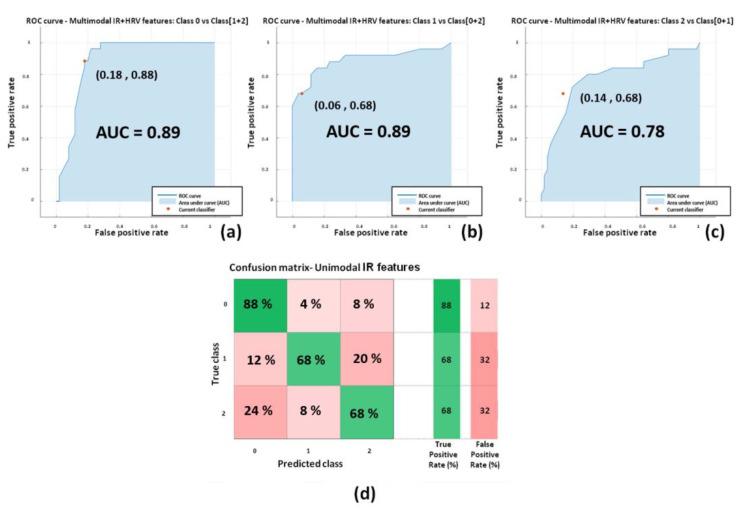
Performances of the Ensemble bagged trees for multimodal IR + hrv features-based classifiers for RAVLT: (**a**) ROC curve for the classifier of class ImmR (i.e., class 0) vs. cumulative class (DelR + Rec) (i.e., class 1 + 2); (**b**) ROC curve for the classifier of class DelR (i.e., class 1) vs. cumulative class (ImmR + Rec) (i.e., class 0 + 2); (**c**) ROC curve for the classifier of class Rec (i.e., class 2) vs. cumulative class (ImmR + DelR) (i.e., class 0 + 1); (**d**) confusion matrix for the multimodal IR features-based classifier.

**Table 1 sensors-22-07300-t001:** Summary of the most related work with research field, measured variables, methodological approach and performances reported.

Authors	Research Field	Measured Variables	Methodological Approach	Performance
Kang et al. [23]	Military training monitoring	thermal IR imaging Modified Cooper Harper Scale ratings Reaction Time	Repeated measure ANOVA Correlation	Nose temperature differs among experimental phases; Significant correlation among all the measured variables r∈ℝ0.88,0.96
Stemberger et al. [24]	Aviator training monitoring	thermal IR imaging cognitive stress test	Repeated measure ANOVA Correlation Artificial Neural Network	Significant change in reaction time as a function of workload level (F(2) = 25.659, *p* < 0.001) Negative relationship between task difficulty andpercentage of correct responses (r(33) = −0.64, *p* < 0.001) 81% correct classification rate
Wang et al. [33]	Thermal comfort and workload indoor	thermal IR imaging EEG environmental thermostat	Repeated measure ANOVA Correlation Random Forest classifier	Average prediction accuracy for all subjects under the slightly cool, neutral, and slightly warm environment is 45% ± 9%, 57% ± 9%, and 44% ± 9%, respectively (prediction of IR features on EEG features) Stronger correlations between absolute facial skin temperature and mental workload are found in the neutral environment, compared to the slightly cool and slightly warm environments.
Or and Duffy [34]	Car driver monitoring	thermal IR imaging Modified Cooper-Harper scale rating	Repeated measure ANOVA Correlation	The workload tasks had no significant effect on forehead temperature Nose temperature showed a significant change after completing tasks for all conditions Significant correlation between the nose skin temperature change and the subjective workload score (r = 0.32, *p* = 0.009)
Pavlidis et al. [32]	Car driver monitoring	thermal IR imaging (perinasal signal to evaluate sympathetic activity) NASA Task Load Index (TLX) steering angle and maximum right-side/left-side lane departure	paired *t*-tests	Mean sympathetic arousal and mean steering performance during cognitive workload had significant deterioration with respect to no-stressor driving (*p* << 0.01)
Perpetuini et al. [12]	Car driver monitoring	thermal IR imaging fNIRS	SVM classifier	Sensitivity of 77% and specificity of 69%

**Table 2 sensors-22-07300-t002:** Participants’ scores statistics.

	DST		RAVLT		
	Forward DST	Backward DST	ImmR	DelR	Rec
**Mean**	7.19	6.15	41.69	8.92	43.08
**Standard Deviation**	2.00	2.13	9.71	3.03	3.74

**Table 3 sensors-22-07300-t003:** Selected features after wrapping method for each of the developed models.

	Unimodal IR Features	Unimodal HRV Features	Multimodal IR + HRV Features
DST	Nosetip: STD; S; SampEn Glabella: K; S; LF; HF	RRmean SDNN HF	Nosetip: STD; S; SampEn Glabella: K; S; LF; HF HRV: RRmean; SDNN; HF
RAVLT	Nosetip: STD; K; S; 90thP; SampEn; LF/HF; LF; HF; Glabella: MeanTemp; K; S; SampEn; LF/HF; LF; HF;	RRmean; RMSSD; LF/HF	Nosetip: STD; K; S; 90thP; SampEn; LF/HF; LF; HF; Glabella: MeanTemp; K; S; SampEn; LF/HF; LF; HF; HRV: RRmean; RMSSD; LF/HF

**Table 4 sensors-22-07300-t004:** Accuracy of the ML classifier for unimodal IR, unimodal HRV and multimodal IR + HRV features-based models. The models with the best accuracy are highlighted in bold.

	Unimodal IR Features		Unimodal HRV Features		Multimodal IR + HRV Features	
	DST	RAVLT	DST	RAVLT	DST	RAVLT
**Decision Tree**						
Simple	46.2	59.2	46.2	44.7	48.1	56.6
Medium	50.0	59.2	50.0	39.5	48.1	53.9
Complex	50.0	59.2	50.	39.5	48.1	53.9
**Discriminant Analysis**						
Linear	69.2	56.6	**59.6**	32.9	63.5	55.3
Quadratic	63.5	53.9	44.2	28.9	59.6	57.9
**Logistic Regression**	69.2	-	50.0	-	**73.1**	-
**Support Vector Machine**						
Linear	**73.1**	65.8	50.0	28.9	**73.1**	63.2
Quadratic	65.4	53.9	42.3	26.3	61.5	51.3
Cubic	51.9	56.6	51.9	35.5	61.5	51.3
**K Nearest Neighbor**						
Coarse	48.1	34.2	48.1	34.2	41.8	34.2
Medium	55.8	47.4	50.0	31.6	57.7	44.7
Fine	59.6	55.3	50.0	44.7	55.8	50.0
**Ensemble**						
Bagged trees	53.8	**71.1**	53.8	44.7	59.6	**75.0**
Subspace discriminant	63.5	56.6	50	32.9	59.6	56.6
Subspace kNN	55.8	56.6	48.1	44.7	57.7	56.6
RUSboosted trees	55.8	47.4	48.1	**47.4**	46.2	35.5

## Data Availability

The data presented in this study are available on request from the corresponding author. The data are not publicly available due to privacy issues. The codes developed for the purpose of the study are available on request to the corresponding author.

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
