# Peer review of "Classification of Drivers’ Mental Workload Levels: Comparison of Machine Learning Methods Based on ECG and Infrared Thermal Signals"

_sensors, 2022, doi:10.3390/s22197300_

Round 1
Reviewer 1 Report
The paper proposes a model based on two cognitive tests (Digit Span Test - and Ray Auditory Verbal Learning Test ) that were administered to participants while driving in a simulated environment, under non-monotonous conditions. The tests were chosen to investigate the drivers' response to predefined levels of cognitive load to categorize the classes of MW. Infrared (IR) thermal imaging concurrently with heart rate variability (HRV) were used to obtain features related to the psychophysiology of the subjects, and to feed machine learning (ML) classifiers. Generally, the topic is interesting but the novelty and motivation should be highlighted compared to previous work. Also, the limitations of previous work need to be mentioned
The abstract The paper's novelty is unclear. Please elaborate and underline the originality in the abstract concerning recent related work.
Introduction:
In line 69, the authors mentioned “IRI”. I guess they meant IR
Again, could you please highlight the novelty and contribution?
Although the authors stated some related studies based on thermal images and ECG, the authors did not mention the limitations of all of these techniques which motivated them to propose a new approach. Please discuss the advantages and limitations of all related works.
Method and Materials.
Could you please mention the inclusion and exclusion criteria for a participant to join the experiment?
Deep learning is the state of the art and since you are using IR images, it was better to use deep learning approaches such as convolutional neural networks. Could you please explain why you did not use deep learning?
Could you please justify the use of feature extraction methods for ECG and IR thermal images? Also, could you please add their equations?
Could you please mention the size of the samples of the dataset and the number of classes? Is the dataset balanced?
It is not clear how you segment ECG signals. Also, did you use any denoising or filtering methods for the ECG signal? Similarly, did you use any enhancement methods for IR images?
There is no need for the section “Application of Supervised Machine Learning for Classification” the description of machine learning and classifier are very known now. You can just mention the classifiers name and method of validation
Experimental Results
What do u mean by “dof” line 310. Also, what is the significance of these results mentioned in this line?
Please dedicate a section to Performance metrics equations and define them. Also, add some other metrics like precision and F1-score
Discussion
Please mention the limitations of your technique.
Conclusion
Please mention your future directions
Reviewer 2 Report
The paper concerns the evaluation of several ML methods and approaches for classifying the mental work load of drivers.It is well-written for the most and easy to follow.
Anyway, I have different concerns that I tried to summarize here below.
Line 77: [...] Significant correlations were found (i.e. r=[0.88,0.96]) [...] Given it seems a range between 0.88 and 0.96, It should be - written in a LaTeX modality - (i.e., $r \in \mathbb{R}[0.88,0.96]$). Shouldn't it?
Introduction: Authors should better highlight the novelties, main contributions and gaps filled in the literature. Furthermore, I recommend a Related work section where they compare point to point (performance, approaches etc) to papers on the same topic. In such a section, a table summarizing primary remarks for every work and this one should be placed.
Participants: authors must provide more data about the participants. For example, what is their experience in driving? How much time do they drive every day? How many times per week? Are they more used to drive in urban contexts or mainly on highways?
Lines 230-242: The paper would benefit from an example of extraction of such features, with actual values and thermal signals. Can be put in the appendix for example.
Lines 255-263: See comment above. It holds true for this case too.
Lines 264-265: what are the baseline values?
Line 274: typo [...] ensemble Classifiers) [...] The parenthesys is not needed. The same typo is present at line 364.
Lines 291-294: please, motivate the choice.
Lines 295-298: Authors state that they used k-fold cross-validation with k=5. I'm not sure this is the correct approach for facing the problem. In studies in which authors tried to classify some human characteristics I see a wide usage of user/human-based validation that ensure the same users data is not present in both validation and training sets. Authors can look at https://doi.org/10.1007/s00521-022-07454-4, for instance. Related to the validation approach, I have another concern: I did not find quantitative information about the dataset. How many samples? How split? And so on...
Lines 361-: unimodal and multimodal approaches refers to two well-known techniques in ML/DL, which are single-view learning and multi-view learning. If unimodal and single-view learning map one on each other perfectly, there are many ways to conduct multi-view learning. The authors should discuss about it comparing with the techniques shown in https://doi.org/10.1007/s00521-022-07454-4. Images are welcome. Moreover, they should motivate better their choices (this is somewhat in touch with my comment on lines 291-294).
Discussion: I suggest to perform an ablation test to check the importance of features.
Overall, for reproducibility: is the dataset available? Is the code available? If not, why?
Round 2
Reviewer 1 Report
The authors have addressed all my comments. I recommend the publication of the manuscript
Reviewer 2 Report
I really appreciate the effort of the authors in revising their paper. They provided exhaustive responses to all my points.
Fantastic work! I hope to read the future works listed in the future.
Good luck with your research.